# Health-Related Quality of Life of Elderly Women with Fall Experiences

**DOI:** 10.3390/ijerph18157804

**Published:** 2021-07-23

**Authors:** Jiyoung Song, Eunwon Lee

**Affiliations:** 1College of Nursing, Korea University, Seoul 02841, Korea; nav857@naver.com; 2Department of Nursing, Gwangju University, Jinwol-dong, Gwangju-si 61743, Korea

**Keywords:** fall, women, health-related quality of life, South Korea

## Abstract

This study aimed to describe the health-related quality of life of elderly women with experience in fall treatment as well as to prepare basic data for the development of interventions to improve the quality of life for this group. The study was based on raw data from the 2019 Korea Community Health Survey. Using the SPSS program, the characteristics of the subjects were tested by frequency, percentage, and chi-square test. To establish the impact of fall experience on the health-related quality of life of elderly women, the OR and 95% CI were calculated using multiple logistic regression analysis. Of the 4260 people surveyed, 44.7% of the elderly women said they had a high quality of life, whereas 55.3% of the elderly women said they had a low quality of life. A younger age was associated with a better-rated health-related quality of life. Those who lived in a city and had a high level of education tended to describe a high quality of life. The quality of life was considered high by those who exercised, but low by those who were obese or diabetic. The results of this study can lead to a better understanding of the experiences of elderly women who have experienced falls, and they can be used as basic data for the development of related health programs.

## 1. Introduction

Falls are commonly occurring accidents in older people, and they have become an important health issue as the number of older people increases with the aging population [1,2]. Over the past year, 15.9% of Korea’s elderly experienced falls, with the fall rate of female seniors recorded at 19.4%, higher than that of male seniors at 11.2%; the hospital utilization rate was also reported to be high [3].

Falls have a wide range of effects from minor accidents to serious injuries and deaths [4], and they also affect the person’s emotional state, degrading the overall quality of life via a loss of confidence and increased social isolation [5,6]. In practice, falls are a predictable and preventable problem in more than 70% of cases; thus, the expansion of social awareness [7] toward the identification of fall risk factors should enable fall prevention arbitration [8]. According to the growth trend of the elderly population, the threat to health and quality of life caused by falls and the socioeconomic costs are expected to further increase in the future; thus, falls could also become a socioeconomic problem [9,10]. Addressing falls in older adults requires a broad understanding of the integrated assessment of various aspects such as balance, depression, and quality of life [11]. Falls are complicated and considered important factors that reduce the quality of life of older adults [5,6].

Health-related Quality of life (HRQoL) can be defined as the subjective satisfaction with life in the physical, psychological, spiritual, and socioeconomic areas [12,13], which becomes more of an issue as life expectancy increases [14]. Factors affecting the health-related quality of life of the elderly include gender, socioeconomic level, physical and mental health, performance of daily life activities, subjective health status, and health behaviors [15,16,17]. In particular, many studies [1,11,18] have found that women’s health-related quality of life is lower than that of men. After the death of the spouse, the elderly woman has a more difficult time than men due to problems such as health, poverty, and depression [18].

Various factors affect the health-related quality of life (HRQoL) of elderly women [11]. It decreases with increasing age, low income, or living alone and low education [17,18]. Additionally, it is low in individuals with chronic diseases, such as high blood pressure and diabetes, depression, and those taking long-term medication [19,20]. Furthermore, HRQoL is low in patients with decreased muscle mass, high BMI, and limited physical activity [15,17], and health behaviors, such as drinking, smoking, and exercise, affect HRQoL [1,18]. Park [12] found that the four variables of depression, physical function, health level, and regular exercise encompass 26% of the variance in women’s HRQoL.

Studies related to health-related quality of life have been conducted in older adults [15,16] and in patients with certain diseases such as diabetes [19,20]. In the current situation in which the elderly population is increasing, it is necessary to learn about the health-related quality of life of elderly women, who are known to experience more falls and have a lower quality of life than men. In this study, we used raw data from the Korea Community Health Survey [21] to examine the health-related quality of life of female senior citizens with experience in fall treatment as well as to prepare basic data for the development of interventions to improve the quality of life.

## 2. Materials and Methods

### 2.1. Study Design and Study Participants

This study was a descriptive research study that used the 2019 Korea Community Health Survey (KCHS) [21] to identify the health-related quality of life according to the fall experiences of elderly women. The KCHS was conducted at 255 health centers nationwide, beginning in 2008, for adults aged 19 or older, to produce comparable regional health statistics to formulate and evaluate regional healthcare plans. In order to collect data, trained investigators visited selected sample households, provided explanations for the investigation and confidentiality to the person surveyed, and received consent to participate. One-to-one interviews were conducted using electronic survey computer-assisted personal interviewing.

Of the 229,099 participants who participated in the 2019 Community Health Survey, elderly women aged ≥65 years with experience of falling treatment were included in this study. The exclusion criteria were men, aged <65 years, no falling treatment experiences, and missing related questionnaire replies (HRQoL, health behaviors, and health-related questions). In this study, fall experience was defined as the presence of fall over the past one year. ‘Have you fallen in the last year (including slips, flips)?’ The single question identified the experience of fall. Finally, a total of 4260 participants were included in the analysis (Figure 1).

### 2.2. Study Variables

#### 2.2.1. Demographic Characteristics

The demographic characteristics of the subjects used data on age (65–69, 70–79, ≥80), area of residence, level of education, and marriage status. The general characteristics were modified to suit the purpose of this study, based on the literature review [8,10,18]. The area of residence was divided into towns and villages. The level of education was categorized as “no studies”, “elementary-school graduation”, “middle-school graduation”, and “high-school graduation”. The marriage status was divided into no spouse (including divorce, bereavement, separation, and unmarried) and having a spouse.

#### 2.2.2. HRQoL

Health-related quality of life was established by the Euro-Qol Group [22] and measured using the Korean version of EQ-5D-3L (EuroQoL-5D), which is widely used to measure quality of life [23]. The EQ-5D index consists of five subindexes: mobility (M), self-care (SC), daily activities (UA), pain and discomfort (PD), and anxiety and depression (AD). In this study, the weighting of the Korea Centers for Disease Control and Prevention (KCDC), which expresses health status as a value from –0.171 to 1 in consideration of the characteristics of Koreans, was applied. The weights were as follows:EQ_5D = 1 − (0.05 + 0.096 × M2 + 0.418 × M3 + 0.046 × SC2 + 0.136 × SC3 + 0.051 × UA2 + 0.208 × UA3 + 0.037 × PD2 + 0.151 × PD3 + 0.043 × AD2 + 0.158A × D3 + 0.05 × N3).(1)

In particular, this study distinguished between high and low quality of life using an EQ-5D index threshold of 60% to determine the health-related quality of life of elderly women with experience in fall treatment.

#### 2.2.3. Health Behavior Characteristics

Health behavior characteristics included smoking, drinking, exercising, and sleeping. Based on the literature review [8,10,18], the health behavior characteristics were modified to suit the purpose of this study. Smoking behavior was categorized into current nonsmokers who had never smoked ordinary cigarettes in their lives, a smoking cessation group who had smoked in the past but did not smoke at the point of data collection, a smoking group who smoked every day, and a smoking group who smoked sometimes. Drinking behavior was categorized into a nondrinking group if alcohol had not been consumed in the past year, or a drinking group if it had been consumed. Exercise was positively defined if it occurred more than once in the past week (days when the body had more than 10 min of intense physical activity leading to more difficult breathing than usual). Sleep was categorized on the basis of the number of hours of sleep per day, divided into groups of 0 to 7 h and more than 8 h.

#### 2.2.4. Physical Health

Physical health characteristics included obesity, hypertension, and diabetes. Obesity was defined as BMI (kg/m^2^) >30; otherwise, BMI was considered normal [18]. Hypertension and diabetes were based on a doctor’s diagnosis [21].

### 2.3. Data Analysis

Data analysis was carried out using the IBM SPSS statistics 23.0 program. Since the community health survey samples were collected under a complex sampling design, a composite sample plan file reflecting stratified variables, colonies, and weights was generated according to the data analysis guidelines of the Korea Disease Control and Prevention Agency, and then, a composite sample analysis was undertaken. The characteristics of demographic factors, health behaviors, and physical health were described using frequency and percentage. Overall differences in proportions between groups were analyzed with the chi-square test. The HRQoL scores according to each variable were presented as the mean ± standard error using *t*-test and ANOVAs. In order to determine the effects of demographics, health behavior, and physical health on the health-related quality of life in older women with experience in fall treatment, multiple logistic regression analysis was conducted. Each odds ratio (OR) was reported together with its 95% confidence interval (CI). All statistical significance levels were set at *p* < 0.05.

### 2.4. Ethical Considerations

The Community Health Survey is a government-designated statistical tool based on Article 17 of the Statistics Act (Approved No. 117015). The Korea Centers for Disease Control and Prevention provide only unidentified data so that individuals cannot be identified from the survey data. The raw data used in this study were requested from the Korea Disease Control and Prevention Agency (https://chs.kdca.go.kr/chs/index.do, accessed on 25 March 2021) and received through the approval process for use. The research was conducted after receiving an exemption from the Public Institutional Review Board of the Ministry of Health and Welfare (IRB No.: P01-202103-22-008).

## 3. Results

### 3.1. Characteristics of Elderly Women According to Health-Related Quality of Life

Group demographic characteristics, physical health, and health behavioral factors are shown in Table 1; 55.3% of the elderly women included in this study (*n* = 4260) considered their quality of life (QoL) to be low. There was a significant difference in age, residence, education, spouse presence, smoking, drinking, exercise, obesity, hypertension, and diabetes between low and high QoL groups (*p* < 0.001); 27.2% of the individuals who reported drinking said their QoL was low, while 24.9% said their QoL was high (*p* = 0.049). In terms of sleep duration, 82.8% and 17.2% of the elderly who slept 0–7 h and >8 h per night, respectively, said they had a high QoL; however, this difference did not reach significance (*p* = 0.053).

### 3.2. HRQoL According to Characteristics of Elderly Women

HRQoL, according to demographic, health behavior, and physical health characteristics of elderly women, is shown in Table 2. The HRQoL score was significantly different as a function of age, living area, education level, spouse presence, smoking, drinking, exercise, sleep, obesity, hypertension, and diabetes (*p* < 0.001). Women who were younger, lived in cities, had a higher level of education, and had a spouse had higher HRQoL scores (*p* < 0.001). Additionally, HRQoL was higher in individuals who did not smoke or drink and regularly exercised. Older women with obesity, diabetes, and hypertension had lower HRQoL scores (*p* < 0.001). In terms of sleep duration, the QoL of elderly women who slept 0–7 h per night was 0.77 ± 0.00, which was higher than in those women who slept >8 h (0.76 ± 0.01; *p* = 0.014).

### 3.3. Factors Affecting Health-Related QoL in Elderly Women

The results of examining the effects of demographic, health behavior, and physical health characteristics of elderly women on health-related quality of life are shown in Table 3.

Compared with women aged 65 to 69, women aged 70 to 79 had 0.51-fold (95% CI: 0.45–0.58) and women aged 80 or older had 0.27-fold (95% CI: 0.23–0.32) lower HRQoL scores. Compared with elderly women living in urban areas, the health-related QoL score of elderly women living in rural areas was 0.81-fold higher (95% CI: 0.74–0.89). At the educational level, the health-related quality of life score for elementary-school graduates was 1.43-fold higher (95% CI: 1.24–1.66), for middle-school graduates was 2.19-fold higher (95% CI: 1.81–2.65), and for high-school graduates was 2.20-fold higher (95% CI: 1.81–2.68) than that of elderly women with no education. In terms of marital status, the health-related quality of life score for elderly women with spouses was 1.04-fold (95% CI: 0.93–1.17) higher than that of elderly women without spouses.

In the case of smoking, the health-related quality of life score of elderly women in the smoking cessation group was 0.58-fold lower (95% CI: 0.42–0.80) and the HRQoL of smokers was 0.72-fold lower (95% CI: 0.52–0.98) than that of nonsmokers. In the case of drinking, the health-related quality of life of the drinking group was 0.94-fold (95% CI: 0.82–1.07) lower compared with the nondrinking group. Elderly women who exercised had a 2.18-fold higher (95% CI: 1.87–2.54) HRQoL score compared with elderly women who did not exercise. With respect to sleep duration, the health-related QoL score of elderly women who sleep more than 8 h was 1.06-fold higher (95% CI: 0.93 to 1.20) than that of elderly women who slept 0 to 7 h.

Compared with nonobese elderly women in this study, the health-related quality of life score of obese elderly women was 0.60-fold lower (95% CI: 0.48–0.75). The health-related quality of life score in elderly women diagnosed with hypertension was 0.90-fold lower (95% CI: 0.81–1.01) than that of undiagnosed elderly women, whereas this proportion was 0.79-fold lower (95% CI: 0.69–0.91) in elderly women diagnosed with diabetes compared with undiagnosed elderly women.

## 4. Discussion

Using raw data from the Korea Community Health Survey, this study sought to determine the quality of life related to health of elderly women with experience in fall treatment. Of the 4260 people surveyed, 44.7% said they had a high quality of life while 55.3% said they had a low quality of life. According to the results of several previous studies [5,6], health-related quality of life is found to be lower in those who experienced falls; however, in this study, health-related quality of life was only evaluated for elderly women with fall treatment experience. In addition, quality of life was distinguished as high or low according to an EQ-5D index threshold of 60%, which will require repeated research in the future.

In the comparison of elderly women with a high quality of life and those with a low quality of life, there were significant differences in all variables except for sleep time. In accordance with the findings of [17,18], quality of life decreased with age. The increase in the number of falls and the degree of injuries with increasing age was not considered, but we were able to identify age as an influential factor on quality of life. Quality of life is accepted as subjective wellbeing and satisfaction with life; the elderly typically exhibit the lowest life satisfaction, warranting increased attention. According to this study, elderly women living in cities had a higher quality of life than elderly women living in rural areas, who exhibited scores that were 0.81-fold lower (95% CI: 0.74–0.89). Studies in other countries have also reported high health-related life scores in socioeconomically developed regions [24]. In this sense, facilities frequently used by senior citizens, such as medical institutions and welfare centers, need to be prepared without regional bias.

In agreement with previous studies [16,24], a higher educational background led to a higher quality of life. It is thought that people with high educational standards perceive a higher quality of life due to their socioeconomic affluence and health activities. As such, continuous education related to hobbies is recommended for senior citizens at welfare centers. In this study, marriage status did not affect the quality of life of elderly women; in another study [25], it was found that marriage status affected the health-related lives of men but not women. Further research is required in this regard.

A previous study of middle-aged women found that smoking had a high influence on the health-related quality of life [26], whereas another study found that smoking did not affect quality of life in older people [18]. In this study, the health-related quality of life of elderly women in the smoking cessation group was 0.58-fold lower (95% CI: 0.42–0.80) and that of elderly smokers was 0.72-fold lower (95% CI: 0.52–0.98) than that of nonsmokers. The number of elderly smokers in this study was small; thus, the generalizability of these results is limited. Additional studies will be needed in the future, in which the entire population can be divided into smoking, smoking cessation, and nonsmoking groups. In terms of drinking alcohol, a previous study mentioned that only drinking volume affects quality of life [27], but this study did not elucidate the factors affecting quality of life. Further studies will be needed to distinguish the amount, frequency, and type of alcohol to establish the impact of drinking on quality of life. In this study, the quality of life of elderly women who exercised was 2.18-fold (95% CI: 1.87–2.54) higher than that of those who did not exercise, in line with previous studies [28,29]. Exercise has the effect of increasing muscle strength to prevent falls, as well as improving quality of life; therefore, it should be actively recommended for elderly women. In addition, it is agreed that frequent walking leads to a higher quality of life, despite it not being a complex form of exercise; thus, various walking programs should be established in the community [1].

In agreement with [1,18], the health-related quality of life of obese women was found to be low in this study. To prevent obesity, in addition to proper exercise, regulated dietary intake and proper eating habits are necessary. It was previously reported that quality of life decreases upon diagnosis of a chronic disease [6,25]. In this study, the quality of life of elderly women diagnosed with diabetes was 0.79-fold lower (95% CI: 0.69–0.91) than that of elderly women not diagnosed. The HRQoL of elderly people with diabetes is directly affected by the dysfunction caused by aging [20], which requires appropriate nursing intervention. It has been reported that a higher number of accompanying diseases leads to a lower quality of life [25]; therefore, in the future, various studies will need to be conducted to consider the type, number, treatment period, and complications of chronic diseases [18].

In this study, we found various factors that affected the HRQoL of elderly women; however, it is important to prevent falls before they occur. Considering these points, it is necessary to improve the socio-physical environment of the elderly and predict the risk of falling by assessing their sense of balance in advance [30]. In addition, research and educational programs should be developed by creating a database of fall risk factors [31].

The investigation of factors affecting the QoL of elderly women with experiences of fall treatment using community health surveys can provide a source of basic data for the development of health-related programs in the future. However, this study has some limitations. It has a cross-sectional design using secondary data, and it is difficult to explain the temporal context of the older women with fall experiences and their related factors. In addition, this study was conducted without considering other diseases, the time of the fall, treatment period, injury area, and surgery of older women in detail. Therefore, further studies are required that would consider the physical, mental health, and environment factors affecting older women.

## 5. Conclusions

This study investigated the factors affecting the health-related quality of life of elderly women with experience of fall treatment using the Korea Community Health Survey. Among 4260 participants, 44.7% considered their quality of life to be high and 55.3% considered their quality of life to be low. Younger respondents had a better health-related quality of life. Those who lived in a city and had a high level of education also considered their quality of life to be high. A high HRQoL was reported by elderly women who exercised, whereas a low HRQoL was reported by obese or diabetic women. This study is fragmentary in that it is not possible to grasp the context of the effect of fall treatment experience on HRQoL. However, it is significant in that it used a community health survey as a representative sample of Korea. The results of this study allow a better understanding of the elderly who have experienced falls and can be used as a source of basic data for the development of related health programs.

## Figures and Tables

**Figure 1 ijerph-18-07804-f001:**
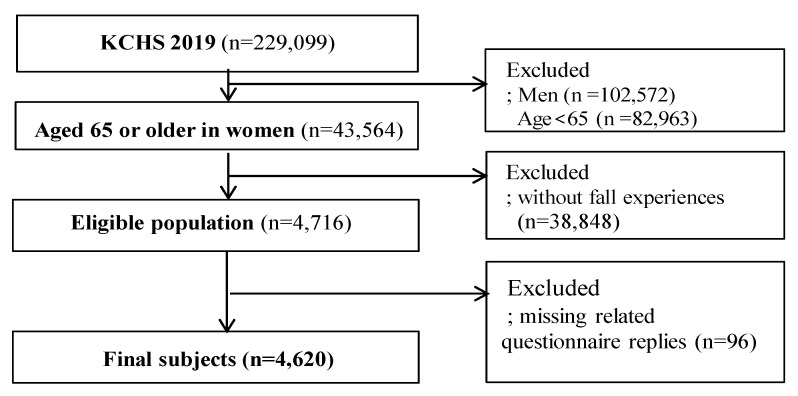
The criteria for inclusion and exclusion of participants.

**Table 1 ijerph-18-07804-t001:** Characteristics of elderly women according to HRQoL (*n* = 4260).

Variables	Categories	Low (*n* = 2355) *n* (%)	High (*n* = 1905) *n* (%)	*p*
Age	65–69	307 (13.9)	594 (33.1)	<0.001
	70–79	1151 (50.9)	968 (50.8)	
	≥80	897 (35.2)	343 (16.1)	
Residence	City	878 (68.7)	858 (76.7)	<0.001
	Town	1477 (31.3)	1047 (23.3)	
Education	No	873 (29.3)	416 (14.2)	<0.001
	Elementary school	1105 (45.2)	894 (41.4)	
	Middle school	216 (12.8)	320 (21.9)	
	≥High school	161 (12.7)	275 (22.5)	
With husband	No	1514 (61.6)	973 (50.1)	<0.001
	Yes	841 (38.4)	932 (49.9)	
Smoking	Nonsmoker	2222 (93.9)	1834 (96.0)	<0.001
	Quit	85 (3.4)	37 (2.0)	
	Yes	48 (2.6)	34 (2.0)	
Drinking	No	1740 (72.8)	1465 (75.1)	0.049
	Yes	615 (27.2)	440 (24.9)	
Exercise	No	2199 (94.9)	1667 (87.3)	<0.001
	Yes	156 (5.1)	238 (12.7)	
Sleep duration (h)	0–7	1854 (81.0)	1521 (82.8)	0.053
	≥8	501 (19.0)	384 (17.2)	
Obese	No	2177 (93.0)	1819 (95.8)	<0.001
	Yes	178 (7.0)	86 (4.2)	
Hypertension	No	824 (35.4)	847 (43.7)	<0.001
	Yes	1531 (64.6)	1058 (56.3)	
DM	No	1732 (70.9)	1498 (77.0)	<0.001
	Yes	623 (29.1)	407 (23.0)	

**Table 2 ijerph-18-07804-t002:** HRQoL according to characteristics of elderly women (*n* = 4260).

Variables	Categories	Mean ± SE	*p*
Age	65–69	0.85 ± 0.00	<0.001
	70–79	0.78 ± 0.00	
	≥80	0.71 ± 0.00	
Residence	City	0.79 ± 0.00	<0.001
	Town	0.76 ± 0.00	
Education	No	0.72 ± 0.01	<0.001
	Elementary school	0.77 ± 0.00	
	Middle school	0.82 ± 0.01	
	≥High school	0.83 ± 0.01	
With husband	No	0.76 ± 0.00	<0.001
	Yes	0.80 ± 0.00	
Smoking	Nonsmoker	0.77 ± 0.00	<0.001
	Quit	0.70 ± 0.01	
	Yes	0.75 ± 0.02	
Drinking	No	0.77 ± 0.00	<0.001
	Yes	0.75 ± 0.01	
Sleep duration (h)	0–7	0.77 ± 0.00	0.014
	≥8	0.76 ± 0.01	
Exercise	No	0.76 ± 0.00	<0.001
	Yes	0.83 ± 0.00	
Obese	No	0.78 ± 0.00	<0.001
	Yes	0.73 ± 0.01	
Hypertension	No	0.80 ± 0.00	<0.001
	Yes	0.76 ± 0.00	
DM	No	0.78 ± 0.00	<0.001
	Yes	0.74 ± 0.01	

Mean ± SE: mean ± standard error.

**Table 3 ijerph-18-07804-t003:** Factors affecting HRQoL in elderly women (*n* = 4260).

Variables	Categories	OR	95% CI
Age	65–69	1	
	70–79	0.51 ***	0.45–0.58
	≥80	0.27 ***	0.23–0.32
Residence	City	1	
	Town	0.81 ***	0.74–0.89
Education	No	1	
	Elementary school	1.43 ***	1.24–1.66
	Middle school	2.19 ***	1.81–2.65
	≥High school	2.20 ***	1.81–2.68
With husband	No	1	
	Yes	1.04	0.93–1.17
Smoking	Nonsmoker	1	
	Quit	0.58 ***	0.42–0.80
	Yes	0.72 ***	0.52–0.98
Drinking	No	1	
	Yes	0.94	0.82–1.07
Exercise	No	1	
	Yes	2.18 ***	1.87–2.54
Sleep duration	0–7	1	
	≥8	1.06	0.93–1.20
Obese	No	1	
	Yes	0.60 ***	0.48–0.75
Hypertension	No	1	
	Yes	0.90	0.80–1.01
DM	No	1	
	Yes	0.79 ***	0.69–0.91

OR: odds ratio; CI: confidence interval; **** p* < 0.001.

## Data Availability

The data presented in this study are available upon request from the Korea Disease Control and Prevention Agency (KDCA). The request for data can be found here: https://chs.kdca.go.kr/chs/index.do (accessed on 25 March 2021).

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
