# Peer review of "Health-Related Quality of Life of Elderly Women with Fall Experiences"

_ijerph, 2021, doi:10.3390/ijerph18157804_

Round 1

Reviewer 1 Report

in the attached document

Author Response

We appreciate your critical review of our work and your suggestions for improving the quality of our manuscript. Based on the comments, we have provided point-by-point responses and have made the associated modifications to the manuscript.

Thank you in advance for your time and attention.

Reviewer 2 Report

Dear Authors,

Congratulations for the work done. However, there are improvements that must be made before your study is eligible for publication:

  1. The Introduction is too short. It should be expanded by developing aspects such as the deterioration of functionality in the geriatric population (both in Korea and in other countries, such as Spain, which is one of the countries with the longest life expectancy in the world; doi: 10.1080 / 02703181.2018.1449163).
  2. Reference should also be made to methods for early identification of balance impairment such as accelerometry (doi: 10.1080 / 21679169.2017.1347707 // 10.1080 / 21679169.2017.1347707). 2. The Discussion should also be slightly expanded by referring to these aspects. 

Kind regards.

Author Response

(The authors gave the same response as above.)

Reviewer 3 Report

In this paper, the authors aim to describe the health-related quality of life of elderly women with experience in fall treatment as well as to prepare basic data for the development of interventions to improve the quality of life for this group. The paper is easy to read and insightful.

Introduction

In the introduction I would add only a few references to underline what was done to assess fall risk (line 14), such as

Maranesi E., et al. The surface electromyographic evaluation of the Functional Reach in elderly subjects. Journal of Electromyography and Kinesiology, 2016, 26, pp. 102–110

Maranesi E. et al. A statistical approach to discriminate between non-fallers, rare fallers and frequent fallers in older adults based on posturographic data. Clinical Biomechanics, 2016, 32, pp. 8–13

Or in diabetic patients (line 28), such as

Maranesi E. et al., Muscle activation patterns related to diabetic neuropathy in elderly subjects: A Functional Reach Test study. Clinical Biomechanics, 2016, 32, pp. 236–240

Materials and Methods

This session is well written and structured.

Results

Insert the meaning of the acronyms under each table.

Table 1 has some problems. In the first row (age), the sum of the subjects of the 3 classes is greater than the total (as well as the sum of the percentages is greater than 100%). The problem of the percentages is present also in the following lines. It is necessary to review the data shown in the Table 1.

Discussion

This session is well structured. I suggest you add a paragraph about the limitations of the study and possible future developments.

Author Response

(The authors gave the same response as above.)

Round 2

Reviewer 1 Report

I would like to thank the authors for their work in the modifications indicated in the first review.

In any case, I would like to make further suggestions to improve the manuscript:

In the description of the statistical analysis I should insist on specifying better which are the statistical tests performed in the comparisons between subjects and variables to establish or not differences and relationships (chi-square, t-student, ANOVA, ...).

In the description of results it is also important that we can know in the text, and in the tables (Table 1, Table 2, Table 3), each p-value to which test applied corresponds.

Author Response

We could see the improvement after revising the paper as you advised. Thank you for your valuable comments to improve our paper. Also, it cheered us up when you mentioned what we did.
We hope you are always healthy even in the COVID-19 era.

As you suggested, we have revised.

Reviewer 2 Report

Dear Authors,

Congratulations on the work done and the changes made to the text.

Kind regards

Author Response

(The authors gave the same response as above.)
